# Internalizing and Externalizing Disorder Levels among Adolescents: Data from Poland

**DOI:** 10.3390/ijerph20032752

**Published:** 2023-02-03

**Authors:** Anna Babicka-Wirkus, Paweł Kozłowski, Łukasz Wirkus, Krzysztof Stasiak

**Affiliations:** 1Institute of Pedagogy, Pomeranian University in Słupsk, Arciszewskiego 22A, 76-200 Słupsk, Poland; 2Institute of Pedagogy, University of Gdańsk, Bażyńskiego 4, 80-309 Gdańsk, Poland; 3Department of Substantive Criminal Law and Criminology, University of Gdańsk, Bażyńskiego 6, 80-309 Gdańsk, Poland

**Keywords:** internalizing disorders, externalizing disorders, adolescents, Poland

## Abstract

This article concerns internalizing and externalizing behaviors among Polish adolescents attending primary schools in a medium-sized city in Poland. The aim of the study was to examine the levels of select problem behaviors (i.e., depression, withdrawal, somatic complaints, aggressive behaviors, delinquent behaviors, thought problems, and internalizing and externalizing disorders) in early adolescence. Another important aim was to establish the ranges of the norm and deviation which would indicate the need for intervention aimed at internalizing and externalizing disorders in the sample. The relationships between variables such as age, gender, and school achievement (grade average) and the groups of problem behaviors and externalizing and internalizing disorders were also examined. To diagnose the occurrence of internalizing and externalizing behaviors, a sample of 550 students (55.3% girls, 46.7% boys) were measured using the Youth Self-Report (YSR) questionnaire. The results showed statistically significant differences in internalizing and externalizing behaviors between boys and girls. Girls achieved higher scores on most of the YSR scales, including internalizing and externalizing disorders, as well as on the total score. The student subgroup scores were also differentiated in terms of age and their average grades. The results also have practical implications; namely, the need for obligatory screening tests of students’ emotional states; encouraging preventive measures in schools, including diagnosis and psychological support in the context of depression; monitoring aggressive behaviors and social problems, both in boys and girls; and implementing universal, selective, and indicated prevention through complex, empirically validated educational-therapeutic programs.

## 1. Introduction

Childhood and adolescence are crucial periods for mental health. Children and adolescents develop cognitive and social-emotional skills which will shape their mental health in the future, and which are important for fulfilling adult roles in society [1]. Numerous significant developmental changes occur in adolescence, especially early adolescence. They involve physical appearance [2,3], personality development [4,5], identity [6,7,8], sexuality [9,10], and developing social competence [11]. Romppanen [11] showed that higher social competence in adolescence is related to a lower number of experienced internalizing problems. Social competence was also correlated with individual styles of coping with stress. Higher social competence co-occurred with more constructive ways of coping with problems [12].

Adolescence is a turbulent developmental period [6], characterized by numerous crises. These crises have several causes, which, according to Thomas M. Achenbach [13], include high occurrence of internalizing and externalizing disorders. Externalizing disorders in children and adolescents involve a high occurrence of destructive and/or aggressive behaviors directed at others and/or the immediate environment. Externalizing disorders include behavioral disorders, attention deficit and hyperactivity disorder, oppositional-defiant disorder, and antisocial personality disorder. On the other hand, internalizing disorders are characterized by emotional problems being experienced internally to a greater extent than they are expressed externally. Internalizing disorders include anxiety disorders, depressive disorders, and obsessive-compulsive disorders occurring in adolescence [14]. The occurrence of these disorders makes it difficult for adolescents to internalize social and legal norms. This frequently results in legal consequences. In Poland, it has been noted that over the past several years, court cases involving adolescents are increasingly concerning violations of social norms (70%) rather than acts punishable by law [15]. In addition, the number of children and adolescents with psychosocial problems to the extent of requiring professional help is similar in many countries and concerns about 10% of their populations. In Poland, this percentage is around 9%. This means that approximately 630,000 children and young people under the age of 18 require the help of a psychological and/or psychiatric support system [16].

The nature of the above relationships has led to increased interest in diagnosing internalizing and externalizing disorders in children and early adolescents in Poland. Because most of the Polish population (around 60%) lives in cities [17], the current study was carried out in a medium-sized (90 thousand citizens) city in which the demographic structure was similar to national trends. This was also the case for the demographics of the study sample (children aged 12–14 years), which comprised around 5% of the population, both in terms of the city’s demographics and the overall Polish demographics.

### 1.1. Theory

T. Achenbach’s concept distinguishes eight dimensions of problem behaviors: (1) withdrawal, or a pathological avoidance of social contact; (2) somatic complaints which occur without any evident cause; (3) anxiety and depression, which may be exhibited through suicidal tendencies or excessive sensitivity to rejection and criticism; (4) social problems, which involve improper functioning in peer groups and not respecting social norms; (5) thought problems, which can be treated as symptoms of mental disorders; (6) attention problems; (7) delinquent behaviors, which involve violations of legal and social norms (e.g., committing punishable acts, school absenteeism); and (8) aggressive behaviors [18,19,20]. Two types of disorders are also distinguished: externalizing (maladaptive behaviors—delinquency and aggression) and internalizing (withdrawal, somatic complaints, anxiety, and depression) disorders, which constitute a typological differentiation of children and adolescents’ conduct disorders [18]. Frequently, adolescents exhibit both externalizing and internalizing behaviors. For example, they may damage property (externalizing behavior) at school and also use drugs or alcohol (internalizing behavior) [21].

Externalizing disorders involve conduct and aggression problems, insufficiently regulated behaviors of an antisocial or oppositional-defiant nature, or behaviors which do not fit within accepted social norms. These all involve projecting internal problems experienced by the individual outwards. The basic symptoms of externalizing disorders are various manifestations of aggression, opposition against one’s surroundings, impulsivity, destructiveness, and antisociality. Their emergence in childhood and adolescence are a significant predictor of chronic criminal behavior in adulthood [22]. Externalizing problem behaviors such as aggression, damaging property, or stealing are among the most frequent adjustment problems in childhood and are the most reliable predictor of mental health problems in adulthood [23]. Children who exhibit externalizing behaviors may suffer a range of legal consequences which could significantly impact their future [24]. High occurrence of externalizing disorders [13] may be a source of social maladjustment. Children and adolescents exhibiting externalizing behaviors also more frequently engage in criminal behaviors and substance abuse in adulthood. Thus, implementing effective prevention of externalizing behaviors in educational programs and services may prevent many problems in psychosocial functioning among adolescents [25,26].

Internalizing behaviors refer to personality problems related to inhibition, anxiety, and overcontrolled behaviors. An excessive sense of control may lead to a deep, neurotic internalization of social norms. This may be the basis of excessive cautiousness in new and subjectively difficult situations, as well as shyness during interpersonal contact. Despite average or above-average intellectual abilities, individuals with internalizing disorders do not achieve adequately high results in school (the so-called inadequate school achievement syndrome), which facilitates a sense of being underappreciated [20]. Anxiety also causes excessively careful following of rules and regulations. Thus, such individuals do not cause problems for others and are usually not perceived as suffering from a disorder by their peers. Most symptoms indicate a passive attitude, but in specific situations (e.g., excessive, frustration-based psychic burden), such individuals exhibit a tendency towards uncontrollable, impulsive behaviors. This may surprise those around them, who then react with excessive punitiveness, leading to even greater withdrawal by individuals suffering from such disorders [22]. The internalizing disorder group comprises emotional and behavioral problems in which experienced problems are introjected inwards and a sense of psychic and somatic discomfort dominates. Four groups of symptoms in children and adolescents’ experiencing internalizing problems are distinguished: (1) anxiety symptoms; (2) depression symptoms (e.g., sadness, low mood); (3) withdrawal, e.g., avoiding social contact; and (4) somatic symptoms without a clear medical cause, e.g., pain symptoms, lack of energy, dizziness, and other somatic complaints [27].

### 1.2. Externalizing and Internalizing Behaviors—Research Review

A child’s understanding of emotions continues to develop up until early adolescence and is coupled with an increase in perspective-taking abilities and a greater understanding of social and moral norms. Therefore, it can be understood as the affective side of social cognition [28]. Children’s behavioral problems in the form of internalizing and externalizing symptoms have been shown to be related not only to social adaptation, but also to the ability to understand others’ emotions [29]. Communicating the experienced psychological and emotional states in relationships with parents and older siblings [30], talking about emotional states [31], socioeconomic status, and attachment quality between family members [32] influence children’s understanding of emotions. The available studies present etiological models of externalizing behaviors (i.e., attachment, withdrawal, and parental attitudes) by testing their relationships with behavioral trajectories from early childhood to adolescence [33].

Behavior problems are frequently long-term. For some individuals, externalizing behaviors are limited to the period of adolescence and its related physiological changes [34]. Internalizing problems are also characterized by anxiety and depression symptoms, social withdrawal, and somatic complaints. During development, gender differences emerge in the frequency of emotional and behavior problems. Boys exhibit higher levels of internalizing symptoms in childhood, whereas for adolescent girls and young women, an increase in internalizing symptoms has been observed, with greater stability in subsequent developmental stages [35]. Long-term consequences of the above relationships include problems in social, school, and future work environments. Studies using confirmatory factor analyses have confirmed the stability and consistency of the eight syndrome model of the YSR, as well as measurement invariance with respect to gender and age [36]. Other studies have showed that gender is a more significant predictor than age and nationality (native population vs. immigrants) [37].

The State of the World’s Children 2021 report, published before this year’s World Mental Health Day, presented an alarming study on children and adolescents’ mental health, highlighting the daily mental health challenges faced by young people, which may lead to disability, diseases, and even death. The report centers on the risk and protective factors in critical life moments, and explores the social determinants of mental health and wellbeing [38]. Early maladaptive experiences at home, in school, or in digital spaces, such as risk of violence, a parent or guardian’s mental disorder, threats, or poverty, increase the risk of developing a mental disorder. Health problems such as epilepsy, developmental disorders, depression, anxiety disorders, and behavior disorders are the main causes of disability among young people. Globally, 10% of children and adolescents experience mental disorders, but most of them do not seek or receive appropriate help. Suicide attempts are an extreme example of a call for help, and they are the fourth most common cause of death for children aged 15–19 [1]. Ivarsson et al. assessed a sample of 237 middle-school students in a small town in Sweden using the YSR. The results showed that four of the YSR scales (withdrawal, anxiety and depression, attention problems, and delinquent behaviors) predicted mild and severe depression. The YSR scales of anxiety and depression and delinquent behavior predicted suicidal thoughts, while the scales of aggressive behaviors and withdrawal (low scores) predicted suicide attempts [39]. This means that the measure used in our study is appropriate not only for diagnosing internalizing and externalizing disorders, but also for drawing conclusions on future problems that adolescents will probably face.

Aside from the problem of demoralization among children and adolescents, the current situation in Poland also points to the necessity of screening young people for mental health problems. In Poland, 9% (630 thousand) of children and adolescents below 18 years of age require professional psychiatric and psychological help. In this respect, Poland does not differ from other countries, where around 10% of children and adolescents also require psychiatric help. However, according to data provided by the National Police Headquarters, suicide is the second most common cause of death among adolescents, and Poland has one of the highest suicide rates in Europe. In the period between 2017 and 2019, out of 1987 suicide attempts, 250 resulted in death. Mental illness was identified as the cause in 585 cases, while mental disorders were identified as causes in 374 cases. The child and adolescent psychiatric care system in Poland does not provide its patients with complex and broadly available support. It promotes a model of environmental therapy for adolescents, though this has not been commonly implemented. This model posits that children should first receive help in a supportive environment (from their family, school, or a specialist clinic) [40]. However, to this end, regular diagnostic screening should be carried out among at-risk groups. The YSR could be broadly used for that purpose.

Children and adolescents’ mental health is a dynamic situation. The period of early adolescence comprises the ages between 12 and 14 years. It is characterized by numerous biological, psychological, and social changes. Change is a definitional feature of this ontogenic period. Moreover, there are significant individual differences in terms of the beginning, duration, and extent of changes experienced during adolescence [41,42]. In adolescence, the turning point, which occurs in early adolescence, is especially important. At this point, potential life scenarios (including the maladaptive and sometimes destructive ones) from childhood have the appropriate conditions for being realized in the process of searching for one’s identity. Adolescence can sometimes be a period of adaptation and improvement in psychosocial functioning. However, due to variable dynamics of change, it can also facilitate the development of various disorders, making it the period of the highest developmental risk [43].

Accordingly, the aim of the current article is to present the results of a study on children and adolescents’ mental health in the period of early adolescence, from the perspective of specific groups of problem behaviors and externalizing and internalizing disorders. Moreover, we also sought to examine the relationships between variables such as age, gender, and school achievement (grade average) and the groups of problem behaviors and externalizing and internalizing disorders. A Polish version of the Youth Self Report questionnaire for adolescents aged 11–18, devised by T. Achenbach, adapted by T. Wolańczyk, was used for this purpose.

## 2. Materials and Methods

### 2.1. Subject of the Study

The subject of the current study was to diagnose the scale of occurrence of behavioral problems in early adolescence. The main research problem in the current study was conceptualized as follows: What is the scale of incidence of behavior problems in early adolescents? The following specific research questions were derived from this research problem:Does gender differentiate the incidence of behavior problems among early adolescents?Does age differentiate the incidence of behavior problems among early adolescents?Does grade average differentiate the incidence of behavior problems among early adolescents?

### 2.2. Study Aims

The first aim of the study was to assess the levels of problem behaviors in early adolescence in specific areas, such as anxiety and depression, withdrawal, somatic complaints, aggressive behaviors, delinquent behaviors, social problems, thought problems, attention problems, and internalizing and externalizing. Regarding the last two areas, it was important to diagnose the normal score range, the cut-off point (indicating the need for psychopedagogical consultation and support), and the clinical score range (indicating the need to assess the relationships between the specific areas of problem behaviors in adolescents and specific variables such as gender, age, and grade average).

### 2.3. Research Tools

To empirically verify the research problem and questions, a Polish version of the Youth Self Report questionnaire for adolescents aged 11–18, devised by T. Achenbach, adapted by T. Wolańczyk was used.

The Youth Self-Report (YSR) is a widely used self-report measure that assesses problem behaviors along two “broadband scales”: internalizing and externalizing. It also allows one to calculate eight empirically based syndromes and DSM-oriented scales and provides a total score. The YSR is a typical “paper-and-pencil” measure. The questionnaire scores are differentiated based on gender (separately for boys and girls) and age (the current study used the version for ages 11–18). The raw scores are transformed into the sten and percentile scales. The percentile scale is divided into three ranges. The “normal” range is located below the 95th percentile, meaning that scores in this area do not qualify the child as having a disorder. The cut-off range is between the 95th and the 98th percentile, and the clinical range extends above the 98th percentile. Children whose scores place in the clinical area are qualified as having a disorder. The answers to the questionnaire questions assess behavior and wellbeing over the last 6 months. The sum of points obtained for individual problem scales, being a raw result, was referred to the modified T-score scale (T factor) to the result for the normative group. This allowed us to categorize each examined person into the healthy group (<61 T), clinical group (>71 T) or borderline group. These values have also been determined in population studies [44].

The YSR is comprised of 112 items, and it measures problem behaviors on eight scales: I—Withdrawal, II—Somatic Complaints, III—Anxiety and Depression, IV—Social Problems, V—Thought Problems, VI—Attention Problems, VII—Delinquent Problems, and VII—Aggressive Behaviors. The total score for the internalizing behaviors scale is obtained by appropriately summing the scores of scales I, II, and III and subtracting the score of Item 103. On the other hand, the total score for the externalizing behaviors scale is obtained by summing the scores of scales VII and VIII [22,32,45,46,47,48].

Statistical analyses were carried out using the IBM SPSS Statistics 25.0 and AMOS 24.0 software.

### 2.4. Statistical Methods

The statistical analyses presented were carried out using the IBM SPSS Statistics 25.0 and AMOS 24.0 software. Basic descriptive statistics together with the test of normality of distribution were calculated for the variables. To compare two groups in terms of the analyzed variables, an independent samples Student’s *t*-test was used. When more than two groups were compared, one-way analysis of variance was used. Next, a regression analysis was carried out to determine the independent variables significant for each of the problem behavior scales. In the final step, a path analysis using the maximum likelihood method was carried out for the relationships between gender, age, grade average, and the YSR total score, as well as the internalizing and externalizing disorder scale scores. The significance level for the analyses was set at α = 0.05.

### 2.5. Research Group

Six hundred and eight students from all of the primary schools in a medium-sized (50–100 thousand citizens) Polish city took part in the study. Due to missing data in some cases, data from 550 participants were used in the analyses. The sample was created by randomly choosing one sixth-, seventh-, and eighth-grade class from each of the primary schools in the city. Thus, the current study involved a total of around 29% of all students from these grades.

In the sample, 55.3% of the participants were girls and 46.7% were boys. The sample comprised 36.4% 12-year-olds, 34.5% 13-year-olds, and 29.1% 14-year-olds. The students’ grade average was mostly between 4 and 5 (the following grading scale is used in Polish schools: excellent—6, very good—5, good—4, satisfactory—3, poor—2 and unsatisfactory—1). A total of 47.8% of the sample had a grade average in this range. The second largest group were the participants with a grade average between 3 and 3.99, which accounted for 26.2% of the sample. Students with a grade average of 5 or higher comprised 21.5% of the total sample. On the other hand, 4.5% of the participants had a grade average of 3 or lower.

Table 1 shows a detailed description of the sample, including frequencies and percentages.

All participants were informed about the study aims and the way the collected data would be used and stored. The students and their parents gave written informed consent to participate. The study’s procedure was carried out in accordance with the Declaration of Helsinki and has received approval from the University Research Ethics Committee (decision no. UKEBN/1/2022).

## 3. Results

The analyses were carried out as described. In the first step, basic descriptive statistics together with the Kolmogorov–Smirnov test of normality of distribution were calculated. The analysis showed that only internalizing disorders had a Gaussian distribution. The remaining variables had distributions which slightly deviated from the normal distribution [49]. Table 2 shows the results of the analyses.

### 3.1. Discriminant Analysis of the Youth Self Report Scales in the Context of Gender, Age, and Grade Average

Girls’ and boys’ YSR scale scores were compared using the independent samples Student’s *t* test. The analysis showed statistically significant gender differences for all the analyzed scales. Girls reported more withdrawal, somatic complaints, anxiety and depression, social problems, thought problems, attention problems, delinquent behaviors, and aggressive behaviors, as well as internalizing and externalizing disorders, and higher YSR total scores than boys. The effect size for these differences was weak to moderate for somatic complaints and anxiety and depression, and strong for internalizing disorders. Greater emotional disinhibition and sensitivity to external stimuli in the girls’ sample may stem from the tendency to somatize the adolescent identity [50] or from stereotypical upbringing patterns which focus on reinforcing women’s emotionality and psychological fragility [51]. Detailed results of the analyses are shown in Table 3.

In the current study, we have shown the intensity of psychological problems among boys and girls in early adolescence, ordered according to T. Achenbach’s concept [18] of eight groups of problem behaviors, as well as externalizing and internalizing disorder scales. The analysis of the collected data showed statistically significant differences between boys and girls in terms of behavior problems in early adolescence (see Figure 1).

Next, a one-way analysis of variance was used to compare the YSR scale results of students of different ages. The analysis showed statistically significant intergroup differences for all YSR scales except thought problems. To assess the character of the differences between the age groups, additional post hoc analyses were carried out using the Bonferroni method when the variances between the groups were homogenous, or the Games-Howell test when they were not. A detailed analysis of the results showed that older students—that is, 13- and 14-year-olds—exhibited higher levels of withdrawal than 12-year-olds (*p* < 0.001). Younger children (12-year-olds) exhibited lower levels of somatic complaints than 13-year-olds (*p* = 0.008) and 14-year-olds (*p* < 0.001). Analogous differences occurred for anxiety and depression—12-year-old students exhibited lower anxiety and depression levels than 13-year-olds (*p* < 0.001) and 14-year-olds (*p* = 0.004). Additionally, 12-year-olds exhibited lower levels of social problems than 13-year-olds (*p* < 0.001) and 14-year-olds (*p* = 0.007). The youngest children in the sample also exhibited lower levels of attention problems than 13- and 14-year-olds (*p* < 0.001). Analogous differences were observed for delinquent behaviors—12-year-old students exhibited lower levels of delinquent behaviors than did older students, including both 13- and 14-year-olds (*p* < 0.001). Regarding aggressive behaviors, lower scores were also observed for 12-year-old students compared to 13-year-olds (*p* = 0.027) and 14-year-olds (*p* = 0.004). Finally, regarding internalizing and externalizing disorders and the YSR total score, lower scores were observed among 12-year-old students compared to 13-year-olds (*p* ≤ 0.007) and 14-year-olds (*p* < 0.001). Thus, it was established that problem behaviors measured by all of the YSR scales except thought problems increase statistically significantly together with the transition into the period of adolescence. The results of the analysis are shown in Table 4.

Next, a one-way analysis of variance was used to compare the YSR scale scores between students with different grade averages. The analysis showed statistically significant intergroup differences in attention problems, delinquent behaviors, aggressive behaviors, and externalizing disorders. To assess the character of the differences between the age groups, additional post hoc analyses were carried out using the Bonferroni method when the variances between the groups were homogenous, or the Games-Howell test when they were not. A detailed analysis of the results showed that students with lower educational achievement also exhibited higher levels of attention problems, delinquent behaviors, and aggressive behaviors compared to students with higher educational achievement (*p* < 0.001). Analogous differences were observed for the externalizing disorders scale. Thus, it was concluded that students with higher grade averages exhibited statistically significantly lower levels of attention problems, delinquent behaviors, aggressive behaviors, and externalizing disorders. The results of the analyses are shown in Table 5.

### 3.2. Regression Analysis of Individual YSR Scales

Next, a regression analysis (see Table 6) was carried out to determine the predictors of each individual group of problem behaviors. The variables included were gender, age, and grade average from one semester prior to data collection.

For the withdrawal scale, two statistically significant predictors were identified; namely, gender, with 2.9% predictive power, and age, which explained 2.4% of the variance. Considering the β coefficient values, it can be concluded that female gender (β = −1.102, *t* = −4.160, *p* < 0.001) and older age (β = 0.506, *t* = 3.807, *p* < 0.001) were predictors of higher withdrawal levels. A similar situation occurred for the somatic complaints scale. In this area, gender was a predictor explaining 7% of the variance, and when considering the β coefficient, it was concluded that female gender (β = −1.853, *t* = −6.493, *p* < 0.001) and (older) age explained 3.2% of the variance (β = 0.638, *t* = 4.383, *p* < 0.001). This relationship was even more clearly evident for the anxiety and depression scale, for which gender was a predictor explaining 10.8% of the variance, while the independent variable of age was responsible for 17.2% of the variance on this scale. The β value for the predictor of gender was 5.071 (*t* = −8.197, *p* < 0.001), and for age, it was 1.057 (*t* = 3.257, *p* < 0.001), meaning that female gender and older age facilitated higher scores on this scale. Regarding the social problems and thought problems scales, it was concluded that no predictors adequately explained their variance. For attention problems, three statistically significant predictors were identified. The most important predictor was gender, which explained 39.6% of the variance. On the other hand, age explained 21.4% of the variance, while grade average explained 4.2% of the variance in attention problems. Considering the β coefficient, it was concluded that female gender (β = −1.306, *t* = −4.858, *p* < 0.001), older age (β = 0.489, *t* = 3.609, *p* < 0.001), and a lower grade average (β = −0.813, *t* = −5.005, *p* < 0.001) facilitated higher scores on this scale. For delinquent behaviors, it was established that gender was not a statistically significant predictor. This correlates with the trend of diminishing differences between boys and girls regarding violations of social and legal norms, which is also reflected in the numbers of court rulings issued in cases involving minors in Poland in recent years [52]. The remaining predictors—that is, age and grade average—explained 5.8% and 5.5% of the variance in delinquent behavior scores, respectively. When evaluating the β coefficient, it was concluded that older age (β = 0.759, *t* = 5.904, *p* < 0.001) and a lower grade average (β = −0.890, *t* = −5.720, *p* < 0.001) facilitated higher scores in this problem area. For the aggressive behaviors scale, three statistically significant predictors were identified—namely, gender, age, and grade average—which explained 4.5%, 3.5%, and 3.2% of variance, respectively. For the aggressive behaviors scale, it was established that female gender (β = −2.662, *t* = −5.212, *p* < 0.001), older age (β = 1.180, *t* = 4.569, *p* < 0.001), and a lower grade average (β = −1.364, *t* = −4.386, *p* < 0.001) facilitated higher levels of this type of problem. The next area analyzed in terms of statistically significant predictors was the internalizing disorders scale. It was established that, in this area, gender was responsible for 14.9% of variance, and age for 2.7%. For internalizing disorders, female gender (β = −9.374, *t* = −9.859, *p* < 0.001) and older age (β = 2.034, *t* = 3.999, *p* < 0.001) facilitated higher scores on this scale. For the externalizing disorders scale, all variables entered into the analysis were statistically significant predictors, with the following predictive power: gender, 3.5%; age, 4.7%; grade average, 3.7%. For externalizing disorders, when considering the β coefficient, female gender (β = −3.134, *t* = −4.545, *p* < 0.001), older age (β = 1.812, *t* = 5.287, *p* < 0.001), and a lower grade average (β = −1.966, *t* = −4, *p* < 0.001) were predictors of higher scores. Finally, for the YSR total score, three statistically significant predictors were identified. They were gender, which explained 4.1% of the variance; age, which was responsible for explaining 4.5% of the variance; and grade average, with a predictive power of 2.1%. For the YSR total score, it was established that the most important predictors were female gender (β = −4.103, *t* = −4.975, *p* < 0.001), older age (β = 2.136, *t* = 5.187, *p* < 0.001), and a lower grade average (β = −1.804, *t* = −3.581, *p* < 0.001).

### 3.3. Path Analysis Using Structural Equation Modeling

To establish the relationships between gender, age, and grade average and the Youth Self Report scales, including the internalizing and externalizing disorder scales, a path analysis was carried out using structural equation modeling with the maximum likelihood method. After considering additional covariance between the variables (based on the modification indices), the analyzed model for the YSR total score achieved a good fit to the data: χ^2^/df = 4.18; CFI = 0.956; RMSEA = 0.076; SRMR = 0.044. Figure 2 shows the standardized regression coefficients for the analyzed model.

The analysis showed a negative association between gender and the YSR total score (β = –0.29; *p* > 0.001). This means that male gender lowered the YSR total score. Similarly, the higher the grade average, the lower the YSR total score (β = –0.18; *p* = 0.002). Age was positively associated with YSR total score (β = 0.19; *p* > 0.001), which means that the older the participants, the higher their YSR scores.

Next, a path analysis for internalizing disorders in the context of gender, age, and grade average was carried out. The analyzed model achieved good fit to the data: χ^2^/df = 3.58; CFI = 0.978; RMSEA = 0.068; SRMR = 0.029. Figure 3 shows the standardized regression coefficients for the analyzed model.

The analysis showed a negative association between gender and internalizing behaviors (β = –0.36; *p* > 0.001). This means that boys exhibited lower levels of internalizing behaviors than girls. For age (β = 0.16; *p* = 0.002), this relationship was negative, meaning that the older the participants were, the higher their internalizing behavior levels. Grade average was not associated with internalizing behaviors (β = –0.08; *p* = 0.069). These results correlate with the results of other studies on Polish children surveyed with the YSR [53].

Next, a path analysis for externalizing behaviors in the context of gender, age, and grade average was carried out. It was established that the model did not achieve a good fit to the data: χ^2^/df = 7.86; CFI = 0.954; RMSEA = 0.112; SRMR = 0.032. Due to insufficient fit to the data, the model was not analyzed.

## 4. Discussion

The results of the current study point to the fact that broadening the perspective on behavior disorders in adolescence to include the area of internalization is warranted. A higher level of internalizing disorders is seen in children who have experienced stressful situations [53,54]. Children and adolescents with clinical results on conduct disorder measures are at greater risk of suicidal thoughts and behaviors than those who score within the norm [55]. As the results show, an important aspect of internalizing behaviors includes anxiety and depression, which are related to a risk of self-harm and suicidal behaviors and are associated with female gender. This is correlated with the statistics on suicide attempts in Poland and deaths by suicide among adolescents. They show that in 2021, police confirmed 1086 suicide attempts among adolescent girls and 410 suicide attempts among adolescent boys. In 2020, these figures were 538 and 305, respectively [56]. According to the World Health Organization, each officially registered death by suicide of an adolescent corresponds to between 100 and 200 suicide attempts.

This conclusion is important for the prevention of mood disorders and presuicidal behaviors from the perspective of including risk factors originating from school (learning difficulties, demands exceeding the student’s capabilities, conflicts, a sense of rejection, criticism of educational achievement), the peer group (social rejection, lack of acceptance, direct and indirect bullying, cyberviolence), the subject, or the adolescent (anxiety disorders, irritability, ASD, destructive mood lability, chronic somatic diseases, neurological conditions, addictions, prolonged internet use, sleep deprivation), and the family (depression in one of the family members, especially the mother; early separation; loss of one or both parents; lack of parental emotional availability; high level of stress in the family; neglect; violence) [57,58]. Other studies verifying the relationship between alexithymia, symptoms of depression, and YSR scale scores among adolescents aged 13–18 have confirmed that in all age groups, girls more frequently exhibited higher levels of depression symptoms than boys [59].

It is worth examining the proportions of scores in the clinical area of the YSR for girls and boys (see Figure 1). Scores in the clinical area predispose the participant to psychological or psychiatric consultation. The first conclusion relates to the fact that for each scale, the proportion of girls who achieved scores in the clinical range was higher than the proportion of boys. The greatest differences were identified for thought problems, in which 33.9% of girls achieved scores in the clinical range compared to 7.4% of boys. According to the cognitive-behavioral model, cognitive distortions lead to inaccuracies and distortions in perceiving and processing data from the surrounding environment. This may lead to inadequate emotional reactions and contextually inappropriate perceptions of behavior [60].

Another troubling conclusion relates to the anxiety and depression scale, on which 30.7% of the girls and 2.7% of the boys in the sample achieved scores in the clinical range. A similar tendency towards higher levels of such emotional problems among girls than boys was also noted by Sadler et al. [61] in a study on boys and girls. Moreover, it bears noting that heightened anxiety (20.5%) and depression (25.2%) among the adolescent population is related to the social-emotional consequences of the COVID-19 pandemic [62].

The third greatest gender difference occurred on the somatic complaints scale, with 17.4% of girls and 5.8% of boys achieving scores in the clinical range. Problems in this area are exhibited through the somatization of tension, causing headaches, stomach aches, dizziness, and nausea due to emotional causes.

Regarding delinquent behaviors, 14% of girls and 2.7% of boys achieved scores in the clinical range. On the social problems scale, 10.6% of girls and 2.3% of boys achieved scores in the clinical range, while for the aggressive behaviors scale, these proportions were 8.9% and 3.1% for boys and girls, respectively.

The smallest gender differences in clinical scores were identified on the attention problems (6.1% of girls and 3.5% of boys) and withdrawal (9.6% of girls and 8.6% of boys) scales. Regarding the externalizing disorders scale, a small difference in clinical scores was observed between girls (3.8%) and boys (3.5%). However, the results were different for the internalizing disorders scale. For this scale, 17.1% of girls and 1.9% of boys achieved scores in the clinical range. These results point to different problem paths exhibited in adolescence by girls and boys. They also point to the need to revise the diagnostic and prevention efforts in schools to emphasize the provision of appropriate programs and specialist consultations to groups of adolescents presenting with homogenous problems.

Based on the analysis of the collected empirical data, no differences between girls and boys on the YSR scales of social problems, thought problems, or delinquent behaviors were identified. Additionally, equal proportions (3.8% and 3.5%) of clinical levels of externalizing disorder scores between boys and girls were observed. When analyzing the characteristic features of this type of disorder related to aggression and maladjustment in adolescence, it is worth considering the cognitive distortions which lay at their core, and which should be targeted by prevention programs in schools. Among adolescents exhibiting tendencies towards aggressive behaviors, four main groups of cognitive distortions can be identified, which serve to rationalize and minimize aversive emotional states. These include egocentrism, blaming others, minimizing the consequences of one’s own actions, and catastrophizing [63]. Additionally, another significant cognitive distortion related to aggression among adolescents—namely, false equivalence—is often highlighted. This involves a belief, common among adolescents exhibiting externalizing behaviors, that other people think similarly to them and hold similar values and beliefs. Aggressive adolescents refuse to accept that others may think and perceive reality in different ways. Emphasizing selected examples of individuals exhibiting aggression leads to the belief that aggressive behaviors are common and that there is nothing wrong with engaging in them. This is also facilitated by a strong aversion towards change, rooted in personal, destructive, or traumatic experiences and the resulting sense of having been harmed [64].

The research project presented in the current article has both strong points and limitations. The strong points include the participation of students from all primary schools in a medium-sized city rather than from a few select ones. As a result, over ¼ of the city’s total student population in the distinguished age groups participated in the study. The participants also frequented schools from each district of the city, which is another strength of the study. The research project also had some limitations. One limitation was the use of only a single measure of problem behaviors. However, this was a result of the adopted model of a diagnostic study. It would be worthwhile to expand the scope of the study by also including adolescents from large cities and small towns, which would facilitate a cross-sectional presentation of this population.

The next stage of the study will involve a deeper analysis using other measures and an expanded study sample, which will include adolescents from other European countries.

## 5. Conclusions

Based on the literature analyzed in the current study, it can be concluded that girls experience more difficulty during the period of adolescence than boys. Higher levels of emotional problems among girls rather than boys are also indicated by the results of other studies [65,66,67]. Girls exhibit greater tendency to express their feelings and emotions through the body than do boys. This is the phenomenon of somatization of the identity being shaped during adolescence [50].

Based on the current results, we posit the following practical implications, the first of which refer to internalizing behaviors. Based on the presented results, it seems warranted to introduce mandatory preventive screening of students’ emotional conditions, with particular emphasis on anxiety and depression, self-harm, and suicidal behaviors in girls. It is also warranted to expand the preventive efforts at schools to include diagnosis of and psychological support related to depression.

Regarding externalizing behaviors, we suggest the following practical conclusion. Problem behaviors in schools are related to social relationships which form the basis for selective and indicated prevention involving both girls and boys. Thus, it is worthwhile to monitor various aggressive and socially maladjusted behaviors exhibited both by boys and girls in schools and implement universal, selective, and indicated prevention through complex, empirically validated educational and therapeutic programs. The need for the implementation of appropriate, evidence-based interventions aimed at children and adolescents’ mental health is also indicated by B. Wright et al. [67]. Additionally, N. Racine et al. [62] highlight the school as an important place for providing emotional support to students.

## Figures and Tables

**Figure 1 ijerph-20-02752-f001:**
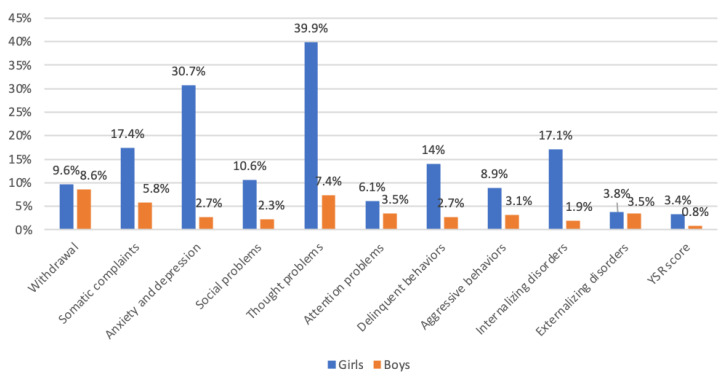
Proportions of clinical range scores on individual problem behavior scales in boys and girls.

**Figure 2 ijerph-20-02752-f002:**
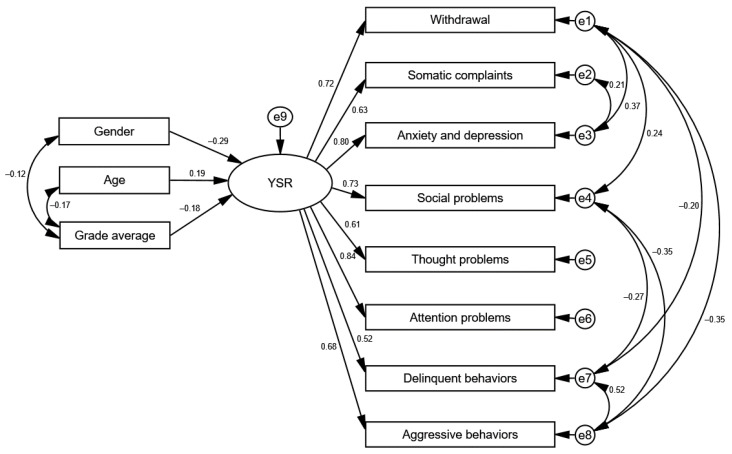
Standardized regression coefficients for the model of relationships between gender, age, and grade average and the YSR total score.

**Figure 3 ijerph-20-02752-f003:**
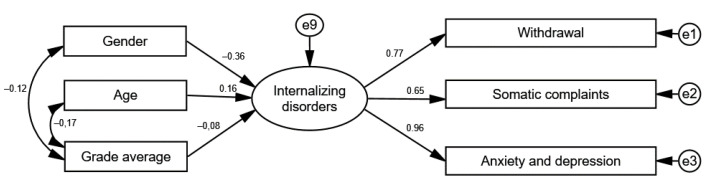
Standardized regression coefficients for the model of relationships between gender, age, and grade average and internalizing behaviors.

**Table 1 ijerph-20-02752-t001:** Demographic characteristics of the sample.

Variables		*n*	%
Gender	Girls	303	55.3
	Boys	257	46.7
Age	12	200	36.4
	13	190	34.5
	14	160	29.1
Grade average	Above 5	118	21.5
	4–5	263	47.8
	3–3.99	144	26.2
	Below 3	25	4.5

**Table 2 ijerph-20-02752-t002:** Descriptive statistics and the distribution normality test.

	*M*	*Me*	*SD*	*Sk.*	*Kurt.*	*Min.*	*Max.*	*D*	*p*
Withdrawal	4.42	400	3.15	0.57	−0.37	0.00	14.00	0.13	<0.001
Somatic complaints	4.24	4.00	3.46	1.10	1.17	0.00	19.00	0.14	<0.001
Anxiety and depression	9.50	8.00	7.10	0.75	−0.14	0.00	31.00	0.12	<0.001
Social problems	3.47	3.00	2.64	0.79	0.29	0.00	14.00	0.14	<0.001
Thought problems	2.89	2.00	2.77	1.01	0.71	0.00	14.00	0.15	<0.001
Attention problems	6.62	6.00	3.21	0.43	0.06	0.00	17.00	0.09	<0.001
Delinquent behaviors	4.82	4.00	3.10	1.02	1.35	0.00	17.00	0.13	<0.001
Aggressive behaviors	10.95	10.00	6.12	0.65	0.14	0.00	34.00	0.10	<0.001
Internalizing disorders	59.00	59.00	12.06	0.03	0.03	24.00	97.00	0.04	0.72
Externalizing disorders	60.52	61.00	8.21	−0.03	0.38	34.00	90.00	0.07	<0.001
YSR score	54.18	55.00	9.86	−0.08	−0.34	24.00	87.00	0.07	<0.001

Note. *M*—mean; *Me*—median; *SD*—standard deviation; *Sk*.—skewness; *Kurt.*—kurtosis; *Min.*—minimum score; *Max.*—maximum score; *D*—Kolmogorov–Smirnov test; *p*—statistical probability.

**Table 3 ijerph-20-02752-t003:** Youth Self Report scale scores in girls and boys.

	Girls (*n =* 293)		Boys (*n =* 257)				95% CI		
	*M*	*SD*	*M*	*SD*	*t*	*p*	*LL*	*UL*	Cohen’s *d*
Withdrawal	4.94	3.22	3.84	2.96	4.16	<0.001	0.58	1.62	0.36
Somatic complaints	5.10	3.62	3.25	2.99	6.57	<0.001	1.30	2.41	0.55
Anxiety and depression	11.75	7.43	6.92	5.71	8.60	<0.001	3.73	5.94	0.72
Social problems	3.70	2.68	3.21	2.58	2.18	0.030	0.05	0.93	0.19
Thought problems	3.13	2.78	2.61	2.73	2.20	0.028	0.06	0.98	0.19
Attention problems	7.23	3.27	5.92	3.01	4.86	<0.001	0.78	1.83	0.42
Delinquent behaviors	5.08	3.18	4.53	2.98	2.11	0.035	0.04	1.07	0.18
Aggressive behaviors	12.19	6.24	9.53	5.65	5.25	<0.001	1.67	3.66	0.45
Internalizing disorders	63.38	11.38	54.01	10.83	9.86	<0.001	7.51	11.24	0.84
Externalizing disorders	61.99	7.94	58.85	8.21	4.54	<0.001	1.78	4.49	0.39
YSR score	56.10	9.78	52.00	9.50	4.98	<0.001	2.48	5.72	0.43

Note. *M*—mean; *SD*—standard deviation; *t*—*t*-test statistic; *p*—statistical significance; *LL* and *UL*—lower and upper confidence interval limit; Cohen’s *d*—effect size

**Table 4 ijerph-20-02752-t004:** Youth Self Report scale scores among 12-, 13-, and 14-year-old students.

	12-Year-Old Students (*n =* 200)		13-Year-Old Students (*n =* 190)		14-Year-Old Students (*n =* 160)				
	*M*	*SD*	*M*	*SD*	*M*	*SD*	*F*	*p*	*η* ^2^
Withdrawal	3.56	2.77	4.97	3.04	4.86	3.47	13.87 ^a^	<0.001	0.04
Somatic complaints	3.46	2.89	4.49	3.78	4.90	3.56	9.95 ^a^	<0.001	0.03
Anxiety and depression	7.70	6.33	10.88	7.44	10.10	7.17	10.96	<0.001	0.04
Social problems	2.89	2.25	3.86	2.74	3.73	2.85	8.92 ^a^	<0.001	0.03
Thought problems	2.52	2.63	3.08	2.94	3.13	2.68	2.88	0.057	0.01
Attention problems	5.62	2.78	7.43	3.16	6.91	3.44	19.36 ^a^	<0.001	0.06
Delinquent behaviors	3.88	2.46	5.05	3.07	5.73	3.51	17.80	<0.001	0.06
Aggressive behaviors	9.82	5.46	11.33	6.10	11.91	6.70	6.15 ^a^	0.002	0.02
Internalizing disorders	55.59	11.28	61.09	12.44	60.79	11.66	13.20	<0.001	0.05
Externalizing disorders	58.61	7.71	61.09	7.89	62.24	8.75	9.66	<0.001	0.03
YSR score	51.23	9.19	55.84	9.91	55.91	9.78	14.81	<0.001	0.05

Note. *M*—mean; *SD*—standard deviation; *F*—ANOVA statistic; *p*—statistical significance; *η*^2^—effect size. ^a^—Welch’s correction was applied.

**Table 5 ijerph-20-02752-t005:** Youth Self Report scale scores for students with different grade averages.

	Below 3 (*n = 25*)		3–3.99 (n = 144)		4–4.99 (*n = 263*)		Above 5 (*n =* 118)				
	*M*	*SD*	*M*	*SD*	*M*	*SD*	*M*	*SD*	*F*	*p*	*η* ^2^
Withdrawal	5.44	3.12	4.59	3.19	4.25	3.06	4.53	3.26	1.93 ^a^	0.103	0,00
Somatic complaints	4.80	4.33	4.29	3.60	4.33	3.61	3.91	2.69	0.88 ^a^	0.476	0,00
Anxiety and depression	10.84	7.31	9.63	7.66	9.46	7.14	9.41	6.18	1.08 ^a^	0.367	0,00
Social problems	4.84	2.48	3.64	2.66	3.30	2.68	3.41	2.50	2.58 ^a^	0.037	0,01
Thought problems	4.08	3.74	2.73	2.67	2.95	2.75	2.74	2.68	1.65 ^a^	0.161	0,00
Attention problems	8.60	3.21	7.14	3.42	6.56	3.09	5.81	2.91	6.81 ^a^	<0.001	0,03
Delinquent behaviors	7.68	4.85	5.42	3.56	4.58	2.56	4.09	2.64	9.70 ^a^	<0.001	0,05
Aggressive behaviors	14.80	7.76	11.58	6.22	11.03	5.98	9.25	5.34	5.55 ^a^	<0.001	0,03
Internalizing disorders	62.72	10.85	58.81	13.19	58.92	12.09	59.45	9.72	2.85 ^a^	0.023	0,02
Externalizing disorders	66.40	10.89	61.44	8.54	60.50	7.37	59.39	8.20	6.49 ^a^	<0.001	0,04
YSR score	59.72	9.93	55.06	10.21	53.97	9.69	52.86	8.88	4.36 ^a^	0.002	0,02

Note. *M*—mean; *SD*—standard deviation; *F*—ANOVA statistic; *p*—statistical significance; *η*^2^—effect size; ^a^—Welch’s correction was applied.

**Table 6 ijerph-20-02752-t006:** Regression analysis of the dependent variables—individual problem behavior groups.

Independent Variable	*R*	*R* Squared	Adjusted *R* Squared	*F*	*p*
Withdrawal					
Gender	0.175	0.031	0.029	17.308	0.000
Age	0.161	0.026	0.024	14.494	0.000
Grade average	0.010	0.001	0.003	0.571	0.116
Somatic complaints					
Gender	0.267	0.071	0.070	42.159	0.000
Age	0.184	0.034	0.032	19.214	0.000
Grade average	0.061	0.004	0.002	2.044	0.153
Anxiety and depression					
Gender	0.331	0.109	0.108	67.204	0.000
Age	0.138	0.189	0.172	10.607	0.000
Grade average	0.446	0.002	0.002	1,093	0.296
Social problems					
Gender	0.093	0.0086	0.007	4.745	0.029
Age	0.109	0.012	0.010	6.709	0.009
Grade average	0.099	0.010	0.008	5.445	0.019
Thought problems					
Gender	0.094	0.088	0.069	4.852	0.028
Age	0.103	0.107	0.089	5.932	0.015
Grade average	0.054	0.003	0.001	1.601	0.205
Attention problems					
Gender	0.203	0.041	0.396	23.602	0.000
Age	0.152	0.023	0.214	13.026	0.000
Grade average	0.209	0.044	0.042	25.053	0.000
Delinquent behaviors					
Gender	0.089	0.008	0.006	4.449	0.035
Age	0.245	0.059	0.058	34.862	0.000
Grade average	0.237	0.056	0.055	32.721	0.000
Aggressive behaviors					
Gender	0.217	0.047	0.045	27.170	0.000
Age	0.193	0.037	0.035	21.128	0.000
Grade average	0.184	0.034	0.032	19.239	0.000
Internalizing disorders					
Gender	0.388	0.151	0.149	97.207	0.000
Age	0.168	0.028	0.027	15.999	0.000
Grade average	0.054	0.003	0.001	1.609	0.205
Externalizing disorders					
Gender	0.191	0.036	0.035	20.655	0.000
Age	0.220	0.049	0.047	27.951	0.000
Grade average	0.198	0.039	0.037	22.287	0.000
YSR total score					
Gender	0.208	0.043	0.041	24.754	0.000
Age	0.216	0.047	0.045	26.910	0.000
Grade average	0.151	0.023	0.021	12.818	0.000

Note. *R*—value of the multiple correlation coefficient; *R* squared—percentage of explained variance; *F-*ratio; *p*—statistical significance.

## Data Availability

Data available on request due to restrictions e.g., privacy or ethical.The data presented in this study are available on request from the corresponding author. The data are not publicly available due to the privacy and professional specificity of the people who took part in this research and who work in the Polish probation system.

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
