# Peer review of "Internalizing and Externalizing Disorder Levels among Adolescents: Data from Poland"

_ijerph, 2023, doi:10.3390/ijerph20032752_

Round 1

Reviewer 1 Report (Previous Reviewer 2)

Thank you very much for giving me the opportunity to review the paper.

I would like to congratulate the authors for their work. As far as I can see in the paper, changes of a previous work have been added to the text. I see more formal flaws that need to be corrected. 

In the case of the introduction, from my point of view, it is too broad, I think that justifying the theoretical framework and the current situation of the problem would be more than enough. 

On the other hand, in the material and methods section, reference is made to the objectives of the study; this should be added at the end of the introduction. 

I don't think it is right to put an explanatory figure in the discussion section either.

And the conclusions section is also too broad.

Author Response

Dear Reviewer,

you will find our responses in the attached file.

Regards,

Authors

Reviewer 2 Report (New Reviewer)

I am concerned that the authors will not take into account the normality of the data, since it is common practice that research using scales generally does not occur with data normality and therefore non-parametric comparative tests should be used, as well as non-linear regression as well.

If the data are not normal, all tests and analyzes must be reflected and this may be reflected in the discussion of the findings.

I suggest corrections in Figure 3, as the line graph is for a time series, and in this case the averages of the dimensions are shown, therefore I suggest a comparative graph (Box & Wiskers) or a graph of mean and standard deviation by gender and dimension, showing the differences.

Author Response

Dear Reviewer,

you will find our responses in the attached file.

Regards,

Authors

Reviewer 3 Report (New Reviewer)

1. The innovation of this study is very small, and only some gender and age differences in mental health have been analyzed, which makes little contribution to the research field of children's mental health.

2. The author's research questions are not clear enough, and the research focus is not prominent enough.

3. 1.1 Part of the authors mainly describe some research data, with little theoretical content..

4.  In the introduction, there is a lack of sorting out the highly relevant literature, and based on these sorting out, the problems of this study are put forward.

Author Response

Dear Reviewer,

you will find our responses in the attached file.

Regards,

Authors

Reviewer 4 Report (New Reviewer)

Line 15 (Abstract): This list of behaviors are components of the” internalizing disorders” dimension, as a matter of fact. Consequently, it is necessary to complement the description of “externalizing disorders” for a due comprehension of what the YSR measures.

Lines 83-84: This information is relevant and should be mentioned in the Abstract, to clarify and rise interest among researchers.

Table 3: The score symbol is not “ P” but “p

Author Response

Dear Reviewer,

you will find our responses in the attached file.

Regards,

Authors

Round 2

Reviewer 2 Report (New Reviewer)

I noticed that the authors included the data normality test and detected what would already be predicted as non-normality due to the use of scales, but this is not a problem, as long as the statistics are well applied. However, in Tables 3 and 4, the use of parametric tests for comparison and groups, if the data are not normal, the tests are incorrect, so I suggest that the authors review this situation. I observed in the method the use of AMOS for analysis of Structural Equations, one more criticism that I make, AMOS is Structural Equations for parametric data, the case reported by the authors is non-parametric data, so AMOS is not appropriate, therefore I suggest the use of SmartPLS or R, and the reading of the following books, and it is up to the authors to review the results found, which in my view should appear divergences depending on the techniques employed by the authors.

HAIR, J. F.; HULT, G. T. M.; RINGLE, C.; SARSTEDT, M. A primer on partial least squares structural equation modeling (PLS-SEM). Los Angeles: Sage publications; 2017.

Hair, J. F. et al. Partial Least Squares Structural Equation Modeling (PLS-SEM) Using R, A Workbook, Open Access, Springer, 2021. https://doi.org/10.1007/978-3-030-80519-7

Author Response

Dear Reviewer,

Thank you for your time and patience. We really appreciate your work.

Statistical analyses were performed using IBM SPSS Statistics 25.0 and AMOS 24.0. Using the program, basic descriptive statistics of the measured variables were calculated together with the test of normality of distribution. In order to compare the two groups in terms of the analyzed variables, an analysis was performed using the student's t test for independent samples. When there were more groups compared, a one-way analysis of variance was performed. In order to determine the relationship between quantitative/order variables, Spearman's correlation analysis was performed. In the last step, path analysis was performed using the maximum likelihood method for the relationship between sociodemographic variables and YSR. The level of significance for the purposes of the analyses was α = 0.05.

We admit that the description of descriptive statistics has been somewhat truncated. However, we believe that due to the volume of the text and its subject matter, this information is not very important for the reader. For large samples, a test of normality is usually relevant. Various simulations show this (see e.g. Noughabi, H. A., & Arghami, N. R. (2011). Monte Carlo comparison of seven normality tests. Journal of Statistical Computation and Simulation, 81(8), 965-972). Therefore, we look at skewness in the second place, and if it is in the range -1;1 or even -2;2 (depending on the source), it is assumed that the distribution is close to the normal distribution (this is what we have in the report from reference to the source - George, Mallery, 2016). Only the skewness information was missing.

Sincerely,

Authors

Reviewer 3 Report (New Reviewer)

This article is too innovative and lacks sufficient academic contributions.

Author Response

Dear Reviewer,

Thank you for your time and patience. We really appreciate your work. However, it is very hard to decide which references are more suitable than others. We did our best to present theory and other study results. In our opinion, our research is very important because of the research group. The study included children from all schools in a medium-sized city, which usually does not happen in other studies.

Sincerely,

Authors

This manuscript is a resubmission of an earlier submission. The following is a list of the peer review reports and author responses from that submission.

Round 1

Reviewer 1 Report

The study aims to evaluate the prevalence of internalizing and externalizing behavior problems among adolescents in a medium-sized city in Poland. Although the results may be of importance from a public health perspective, the manuscript itself lacks clarity and is difficult to follow. For example, some parts of the text describing ithe background and aims of the study are placed under Methods, the Results section is absent, the results are described under Discussion, and a short discussion of the study results is under Conclusions. Of note, there are two Conclusions sections. It is also difficult to note a common thread through the manuscript due to many different ideas/theories as well as often superfluous details. Thus, my suggestion for the authors is to give an additional thought on what is the main message they want to convey to the reader and revise the manuscript according to that.

Reviewer 2 Report

Thank you very much for giving me the opportunity to review this paper. First of all I would like to congratulate the authors. I think it is a very interesting and accurate paper. 

From my point of view, the structure of the article could be improved.

In the introduction, from line 59 to 68, last paragraph, it does not correspond to the introduction, where a current status of the situation and justification of the work should be given.

Point 3 on theory should be deleted. Its content should be partly in the introduction, current state of the question, and partly in material and methods, justifying the tool. 

There should be an exclusive section on results to present the discussion afterwards, not mixed in the section. 

Reviewer 3 Report

Based on adolescent sample this paper aims to to assess the levels of internalizing and externalizing disorders in adolescence take to account variables such as gender, age, and the grade average. I recommend to prepare a major revision of the manuscript based on my notes and to re-submit it.

---Specific comments---

-        Abstract is insufficiently described. Authors should include in this section at least the study aim, the sample used (e.g., n, Mean, SD, % girls and % boys…), the statistical analyzes carried out and the conclusions and/or practical implications.

-        The Introduction section is underdevelopment needs revision. I consider that there is an important point is missing in the literature review that would add value to this section. This is a literature review focused on gender, age and grade average. Currently almost all the studies cited focus on adolescent in general but the sample used is young adolescent population and authors analyse the gender, age and grave average differences, so, these variables should be included in the Introduction section. Finally, I recommend to add the section tittled “Theory” to Introduccion section.

-        For the measure that were used, please add an item example and indicate whether response scores were summed or averaged to create their composite scores for data analysis.

-        Create a table would be useful in order to describe the characteristics of the sample.

-        The Results section is missing. I believe the authors confuse the results section with the discussion section and conclusión section.

-        Overall, the Discussion section is quite brief and underdeveloped. This section should be start with the study aim. There is no sense to include in this section the program used (SPSS and AMOS) or the statistical analysis carried out, this information should be included in the results section. In this section authors should discuss and explain the finding. I suggest the authors review the entire section and make an effort to connect it with the information provided throughout the manuscript according to the changes that I suggest in previous sections. This section ends with strengths and limitations of the study.

-        The conclusion section should be included the principal knowledge add to the study area and the implications of the study.

-        Please, revise the entire manuscript according to 7th Edition APA style.